

# The reporting of $p$ values, confidence intervals and statistical significance in Preventive Veterinary Medicine (1997–2017)

Locksley L. McV. Messam[1,*], Hsin-Yi Weng[2,*],
Nicole W. Y. Rosenberger[1], Zhi Hao Tan[1], Stephanie D. M. Payet[1] and
Mahishi Santbakshsing[1]

[1] Section: Herd Health and Animal Husbandry, University College Dublin, School of Veterinary Medicine, Dublin, Leinster, Ireland
[2] Department of Comparative Pathobiology, College of Veterinary Medicine, Purdue University, West Lafayette, Indiana, USA
* These authors contributed equally to this work.

Corresponding author
Locksley L. McV. Messam,
locksley.messam@ucd.ie

## ABSTRACT

**Background:** Despite much discussion in the epidemiologic literature surrounding the use of null hypothesis significance testing (NHST) for inferences, the reporting practices of veterinary researchers have not been examined. We conducted a survey of articles published in Preventive Veterinary Medicine, a leading veterinary epidemiology journal, aimed at (a) estimating the frequency of reporting $p$ values, confidence intervals and statistical significance between 1997 and 2017, (b) determining whether this varies by article section and (c) determining whether this varies over time.

**Methods:** We used systematic cluster sampling to select 985 original research articles from issues published in March, June, September and December of each year of the study period. Using the survey data analysis menu in Stata, we estimated overall and yearly proportions of article sections (abstracts, results-texts, results-tables and discussions) reporting $p$ values, confidence intervals and statistical significance. Additionally, we estimated the proportion of $p$ values less than 0.05 reported in each section, the proportion of article sections in which $p$ values were reported as inequalities, and the proportion of article sections in which confidence intervals were interpreted as if they were significance tests. Finally, we used Generalised Estimating Equations to estimate prevalence odds ratios and 95% confidence intervals, comparing the occurrence of each of the above-mentioned reporting elements in one article section relative to another.

**Results:** Over the 20-year period, for every 100 published manuscripts, 31 abstracts (95% CI [28–35]), 65 results-texts (95% CI [61–68]), 23 sets of results-tables (95% CI [20–27]) and 59 discussion sections (95% CI [56–63]) reported statistical significance at least once. Only in the case of results-tables, were the numbers reporting $p$ values (48; 95% CI [44–51]), and confidence intervals (44; 95% CI [41–48]) higher than those reporting statistical significance. We also found that a substantial proportion of $p$ values were reported as inequalities and most were less than 0.05. The odds of a $p$ value being less than 0.05 (OR = 4.5; 95% CI [2.3–9.0]) or being reported as an inequality (OR = 3.2; 95% CI [1.3–7.6]) was higher in the

abstracts than in the results-texts. Additionally, when confidence intervals were interpreted, on most occasions they were used as surrogates for significance tests. Overall, no time trends in reporting were observed for any of the three reporting elements over the study period.

**Conclusions:** Despite the availability of superior approaches to statistical inference and abundant criticism of its use in the epidemiologic literature, NHST is substantially the most common means of inference in articles published in Preventive Veterinary Medicine. This pattern has not changed substantially between 1997 and 2017.

## INTRODUCTION

In March 2016, The American Statistical Association (ASA) issued a statement on the use of *p*-values and statistical significance in scientific articles (*Wasserstein & Lazar, 2016*). This was in response to continued and widespread misuse of null hypothesis significance testing (NHST) in the quantitative sciences (*Wasserstein & Lazar, 2016*). NHST is typically characterised by the use of *p* values as a cut-off (usually 0.05) for distinguishing between scientifically important and non-important results, notwithstanding the lack of a firm scientific basis for this (*Gill, 1999*; *Goodman, 2008*; *Goodman, 2016*; *Rothman, 1978*; *Sterne & Davey Smith, 2001*). Consequently, *p* values are often incorrectly reported as inequalities (*e.g.,* "$p < 0.05$" or "$p < 0.01$") (*Goodman, 2008*), reinforcing the erroneous impression that their precise magnitude is irrelevant and that use of the particular cut-off has merit. These practices ultimately degrade inferences based on estimated parameters into (statistically) significant and non-significant categories without any consideration of the magnitudes of the estimates themselves (*Nuzzo, 2014*), and how they compare to previous estimates obtained by other researchers (*Poole et al., 2003*). Increasingly, over the last 40 years, authors from various fields have commented on, and criticised the afore-mentioned practices (*Fidler et al., 2004a*; *Gill, 1999*; *Goodman, 2008*; *Goodman, 1999*; *Johnson, 1999*; *Nuzzo, 2014*; *Rothman, 1978*; *Rothman, 1986*; *Savitz, 1993*; *Sterne & Davey Smith, 2001*; *Trafimow, 2014*; *Trafimow & Marks, 2015*; *Utts, 1988*) and attempts have been made to improve statistical inference and reporting in various disciplines (*Fidler et al., 2004a*; *Gardner & Altman, 1988*; *International Committee of Medical Journal Editors, 1988*; *Wilkinson, 1999*).

Within the discipline of epidemiology, there has been notable concern and objection to the use of NHST, with editorials and commentaries published in a number of sub-speciality journals (*Feinstein, 1998*; *Lang, Rothman & Cann, 1998*; *Lash, 2017*; *Poole, 1987*; *Stang, Poole & Kuss, 2010*) as well as attempts made to quantify its use and impact on the field (*Fidler et al., 2004b*; *Holman et al., 2001*; *Perneger & Combescure, 2017*; *Pocock et al., 2004*; *Savitz, Tolo & Poole, 1994*; *Stang et al., 2017*). In the past, one major epidemiology journal strongly discouraged the use of *p* values (*Lang, Rothman & Cann,*

*1998*) and most recently, an epidemiologist co-authored a call for abandonment of the use of the concept of statistical significance (*Amrhein, Greenland & McShane, 2019*). Specifically, they asked for the discontinuation of the conventional application of the *p* value = 0.05 as a bright-line test for whether a particular result supports or refutes a scientific hypothesis. This request was endorsed by over 800 fellow scientists (*Amrhein, Greenland & McShane, 2019*).

Confidence interval use in the biomedical sciences has also been criticised because of its analogous weaknesses to those of *p* values: The habitual 95% confidence level is similarly arbitrary and without a sound basis. Their correspondence with *p* values is often exploited to subtly indicate whether a result is statistically significant or not, resulting in the same inferential shortcomings (*Feinstein, 1998*). Additionally, unlike Bayesian credible intervals, confidence intervals cannot be used to express the probability that the estimated parameter is located within its limits. Epidemiologists, while critical of this misuse of confidence intervals, have nevertheless tended to encourage their reporting alongside *p* values (*Witte, Thomas & Langholz, 1995*) or in lieu of them, correctly emphasising that confidence intervals can provide all information that *p* values provide, and more importantly, are useful as means of conveying the precision of estimated parameters (*Naimi & Whitcomb, 2020*; *Lang, Rothman & Cann, 1998*; *Poole, 2001*). While this latter is not true in all cases (*Morey et al., 2016*), for common epidemiologic measures (odds, risk and rate ratios, risk and rate differences) the width of the confidence interval is directly proportional to the standard error of the estimate and is therefore a legitimate measure of its precision (*Greenland et al., 2016*; *Naimi & Whitcomb, 2020*). Confidence intervals also communicate the precision of an estimate in a manner distinct from its magnitude. This is in contrast to the *p* value, which is dependent on both the magnitude and precision of the estimate in a manner that the reader cannot disentangle. Yet a more fundamental problem in using a *p*-value's magnitude for inference in disciplines where parameter measurement is important, is that unlike confidence intervals, it does not provide an answer to research questions in the metric of the parameter of interest. This point is not new (*Poole, Lanes & Rothman, 1984*) and was alluded to in the ASA's statement (*Wasserstein & Lazar, 2016*).

Notwithstanding attention paid to NHST by epidemiologists in human-related fields, veterinary epidemiologists have been silent on the matter. While it may be postulated that NHST is as widespread in veterinary medical research as it is in other fields, we have not found evidence of any attempts to document this in the veterinary epidemiologic literature or, for that matter, any commentaries suggesting that this is an area of concern to veterinary epidemiologists.

The goal of this project is two-fold. First, to report on the frequency of use of the reporting elements: *p* values, confidence intervals and statistical significance for inference in articles published in Preventive Veterinary Medicine, a leading journal of veterinary epidemiology. Second, to determine if the use of these reporting elements varies between (a) article sections and (b) over time. While investigating this, we consider the form in which *p* values are reported, the use and interpretation of confidence intervals, as well as the context in which statistical significance is used. We focus primarily on the patterns and
frequency of use of these reporting elements in each article section as a way of documenting the relative importance of NHST to inferences communicated in different parts of a manuscript, as well as in the manuscript overall. We examine publications from 1997 to 2017 in order to identify any notable secular trends in reporting and to include substantial portions of the previous three decades in which there has been notably increased commentary and major published empirical research on the use of NHST by epidemiologists (*Chavalarias et al., 2016*; *Cristea & Ioannidis, 2018*; *Fidler et al., 2004b*; *Greenland et al., 2016*; *Holman et al., 2001*; *Lang, Rothman & Cann, 1998*; *Lash, 2017*; *Poole, 2001*; *Rothman, Greenland & Lash, 2008*; *Savitz, Tolo & Poole, 1994*; *Stang et al., 2017*; *Stang, Poole & Kuss, 2010*; *Wasserstein & Lazar, 2016*).

## MATERIALS AND METHODS

### Sampling frame

The sampling frame for the study consisted of all original research articles published in Preventive Veterinary Medicine (https://www.journals.elsevier.com/preventive-veterinary-medicine) from January 1, 1997 to September 30, 2017 inclusive.

### Sampling

We applied a 1-in-3 systematic cluster sampling approach with each cluster being a month of publication. First, we randomly selected 1 month from among the first 3 months of the year (January to March) and thereafter selected every third month for study inclusion. March was randomly selected as the starting point and thus, for each year of the study period, we sampled all original articles published in March, June, September and December. When for a given year, there was no journal issue in one of the chosen months, the preceding month was selected for study inclusion.

### Search procedure

Articles within sampled clusters were downloaded in pdf format prior to being searched. Every article's title, year and month of publication and volume number were noted. Each article was then searched using the following search terms: "*p*" for *p* values, "CI C.I. CL C.L. confidence" for confidence intervals and "significance significant significantly" for statistical significance. For searching, we used the "Full Reader Search" function in Adobe Reader DC version 10 along with the "Match Any of the words" and "Whole words only" sub-menus. In each article, we searched the abstract, results-text, results-table(s) (including footnotes), and discussion for one example of each reporting element without regard to the presence of the others. Each search result was binary (*i.e.* Yes/No). If results-texts and discussions were written as one section, the search result was recorded separately under both sections. Searches were performed by co-authors MS, SP, NR and ZHT for March, June, September and December publications, respectively.

### Detailed review of reporting practices

Based on initial search results, a subsample consisting of articles in which the reporting elements were found, was selected for detailed review. The articles were divided into 24

strata defined jointly by the reporting element (singly: $p$ values only, confidence intervals only etc. or in combination: $p$ values and confidence intervals, confidence intervals and statistical significance etc.) and the article section (*e.g.*, abstract, results-text, etc.) containing them. We excluded articles with Bayesian analyses but no frequentist content and articles in which $p$ values and confidence intervals were reported as part of model selection or model fitting criteria, except when the model selection or model fitting was the study objective.

From strata containing thirty or more articles, we randomly sampled 20 percent for further review. When the number of articles in a stratum was less than 30, five or all (if less than five) articles were chosen. Random samples were generated using the IBM SPSS Statistics for Windows (Version 24.0; IBM Corp., Armonk, NY, USA) Random Number Generator with fixed seeds. Each article was independently reviewed by both LM and HYW with discrepancies resolved by consensus following discussion.

### Review of the form in which $p$ values are reported

For each article section in which a $p$ value was reported, we noted if the exact value ($p =…$) or an inequality ($p <, >, \geq,$ or $\leq…$) was written. When an inequality involving a number less than or equal to 0.001 (*e.g.*, $p < 0.001$) was reported, this was considered the exact value, as we assumed the reporting of the inequality was likely a result of the limitations of the statistical programme. Additionally, in the abstracts, results-texts and results-tables we counted the number of $p$ values with magnitudes (a) less than and (b) greater than or equal to 0.05. Whenever "$p <$" or "$p \leq$" was reported along with a number greater than 0.05 (*e.g.*, "$p < 0.2$"), this was considered greater than or equal to 0.05. To avoid duplication, $p$ values reported in results-texts that were used to refer to $p$ values in results-tables were not counted.

### Review of the usage and interpretation of confidence intervals

Where confidence intervals were reported in an article section, we noted (a) whether or not NHST was the analytic goal; (b) if any of the reported confidence intervals were interpreted; and if so, (c) what proportion were interpreted as if they were used to perform a significance test. Results were categorised as all, some or none of them. We deemed that a confidence interval was used to perform a significance test, if the accompanying point estimate was described as "significant", "statistically significant" (or their negatives) in the absence of a $p$ value, or it was stated whether it included the null value (*e.g.*, 1, for ratio measures).

We also noted whether the confidence intervals were reported for only commonly used epidemiologic measures, only for other measures, or for both. We considered the following, common epidemiologic measures: Measures of disease occurrence (incidence rates and proportions, prevalence, and mortality), measures of association (odds, risk, rate and prevalence ratios and risk differences), sensitivity, specificity, and median survival time.

### Review of the reporting of statistical significance

Whenever "significant", "significance" or "significantly" (hereafter "significant…") were found in a section, we noted whether they were used to indicate statistically significant study results, statistically non-significant study results or both. Statistically non-significant results were considered those which were described by a sentence expressing the converse of statistical significance. Whenever "significant…" did not refer to study results, they were excluded from consideration.

### Data quality verification

During detailed review of articles, we noted and corrected all occasions in which the appearance of a reporting element was erroneously recorded (false positives). To verify the absence of each of the three reporting elements, we randomly selected 30 sections from each of the four article sections that did not contain the respective reporting element (thus $12 \times 30 = 360$ article sections in total). For instance, to verify the absence of $p$ values from abstracts, we randomly chose a sample of 30 abstracts from among all abstracts that were found to not contain any $p$ values during the initial search. We noted all occasions in which the absence of a reporting element was erroneously recorded (false negatives) during the initial search and corrected the errors. Verification was done by LM and HYW, by repeating the initial search procedure used to identify reporting elements.

## DATA ANALYSIS

### Estimation of article section-specific proportions

The unit of analysis was the individual article section (abstract, results-text, results- table(s) or discussion). Data were analysed in Stata Statistical Software (Release 14; StataCorp LP., College Station, TX, USA). Overall and annual estimates along with associated 95% confidence intervals (CIs) for proportions of article sections containing each combination of the reporting elements were calculated using the survey data analysis menu. We then plotted article-section specific yearly point estimates to visualize any trends over the study period.

### Detailed review

#### P values

For abstracts, results-texts, results-tables and discussions in which $p$ values were reported, we estimated the proportions in which all, some and none of the $p$ values were written as inequalities as opposed to in exact form. Weighted averages of the proportion of $p$ values that were less than 0.05 were also estimated for each section.

#### Confidence intervals

For abstracts, results-texts and discussions in which confidence intervals were reported, we estimated the proportions in which the confidence intervals were reported for only the commonly used epidemiologic measures. Additionally, for the subset of these articles in which NHST was possible, we also estimated the proportion of abstracts, results-texts and

discussions in which all, some and none of the confidence intervals were interpreted as significance tests.

### Statistical significance

For abstracts and discussions in which "significant…" referred to study results, we estimated the proportions of the sections in which the term was used to refer to (a) only statistical significant results, (b) only statistically non-significant results and (c) both.

### Comparisons between sections

Finally, we estimated prevalence odds ratios (ORs) and 95% CIs comparing the odds of a given reporting element occurring in one section relative to another section. For this we used Generalised Estimating Equations with a logit link function, binomial distribution, and exchangeable correlation structure (*Twisk, 2003*), to account for the dependency of sections within articles.

## RESULTS

Nine-hundred and eighty-five articles were searched initially. Of these, 839 (85%) had an abstract, results-text, results-table, and/or discussion section containing at least one reporting element. Four hundred and fifty-seven article sections containing different combinations of reporting elements were used for the detailed review of sections containing the reporting elements. Average false positive and negative errors for the three reporting elements across the four sections during initial searches were 5 and 1.5%, respectively.

### Overall reporting

Over the study period, the proportions of abstracts (31%; 95% CI [28–35]), results-texts (65%; 95% CI [61–68]), results-tables (23%; 95% CI [20–27]) and discussions (59%; 95% CI [56–63]) reporting "significant…" exceeded both those proportions reporting $p$ values and those reporting confidence intervals in those sections except in results-tables (Fig. 1). The proportion of sections reporting confidence intervals and $p$ values were similar, except among results-texts where the proportion reporting $p$ values was substantially higher (Fig. 1). The proportion of results-texts containing $p$ values and "significant…" (separately) were higher than the proportion of other article sections containing these reporting elements (Fig. 1). Confidence intervals were reported in a substantially higher proportion of results-tables (44%; 95% CI [41–48%]) than in other sections (Fig. 1). Overall, the odds of reporting "significant…" was substantially higher than both the odds of reporting confidence intervals and the odds of reporting $p$ values, respectively, in abstracts, results-texts and discussions (Fig. 2). Only in results-tables were odds of reporting $p$ values or confidence intervals higher than the odds of reporting statistical significance (Fig. 2). The odds of reporting $p$ values were roughly equal to the odds of reporting confidence intervals in all sections except results-texts where they were higher (OR = 2.5; 95% CI [2.1–2.9]) (Fig. 2).

In each section, there was substantial yearly fluctuation in the proportions containing reporting elements over the study period, with average yearly changes of at least

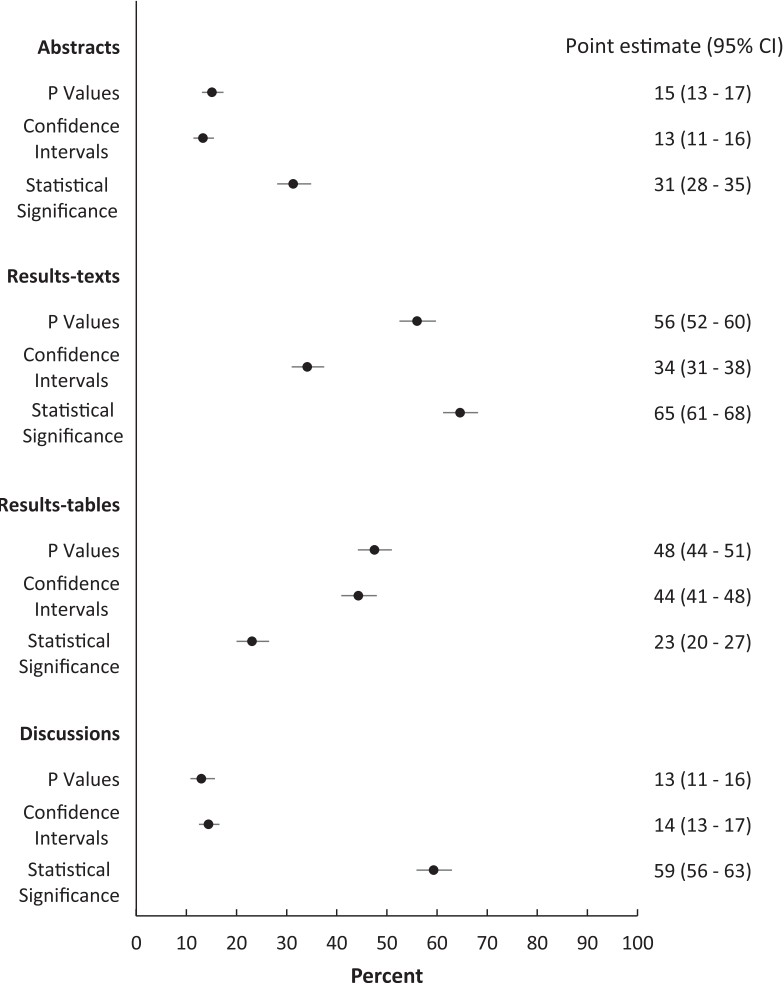

**Figure 1** **Proportions (%) of abstracts, results-texts, results-tables and discussion sections reporting *p* values, confidence intervals or instances of statistical significance.** Point estimates and 95% confidence intervals (CIs) for the proportions (%) of abstracts, results-texts, results-tables and discussion sections reporting at least one *p* value, confidence interval or instance of statistical significance (regardless of the presence of the others) in original research articles published in Preventive Veterinary Medicine (1997–2017). In each abstract, result-text, results-table and discussion, "*p*", "CI C.I. CL C.L. confidence" and "significance significant significantly" indicated the presence of *p* values, confidence intervals and statistical significance, respectively.

5% (Figs. 3A–3D). Nevertheless, no section showed any net tendency towards either increases or decreases in reporting *p* values, confidence intervals or "significant…" over the study period. Overall, yearly proportions of results-tables reporting "significant…" were consistently higher than those reporting *p* values and confidence intervals in both the abstracts (Fig. 3A) and discussions (Fig. 3D).

Among sections in which only one reporting element was present, the pattern was similar to the above overall pattern (Fig. 4), with the proportions in sections reporting only "significant…" exceeding proportions of sections reporting only *p* values and only confidence intervals except among results-tables (Fig. 4).

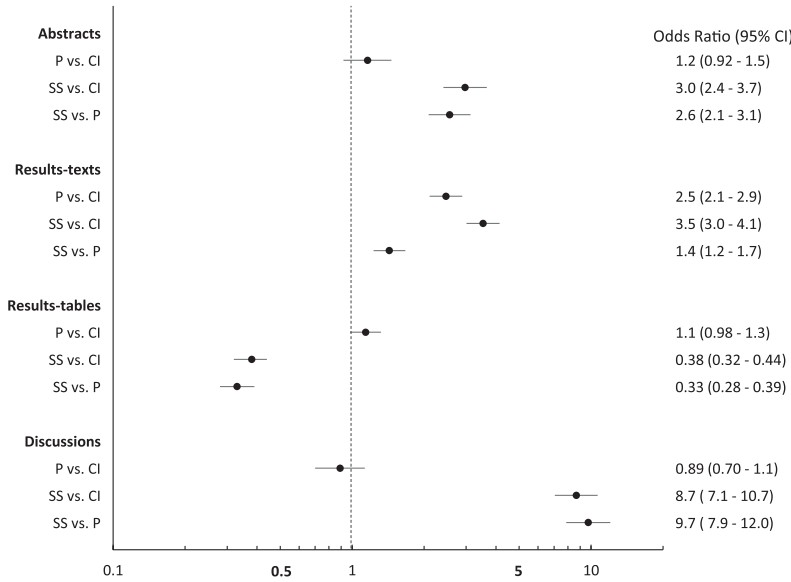

**Figure 2 Odds ratios and 95% confidence intervals comparing abstracts, results-texts, results-tables and discussion sections reporting *p* values, confidence intervals or statistical significance.** Odds ratios and 95% confidence intervals (CIs) for the comparison of the proportions of abstracts, results-texts, results-tables and discussion sections reporting at least one *p* value (P), confidence interval (CI) or instance of statistical significance (SS) (regardless of the presence of the others) in original research articles published in Preventive Veterinary Medicine (1997–2017).

When combinations of the reporting elements were investigated, the proportion of abstracts, results-texts and discussions reporting *p* values and "significant…" together, were higher than the proportion reporting other combinations in corresponding sections (Fig. 4). Among results-tables, the proportion reporting *p* values and confidence intervals together was higher than those reporting other combinations (Fig. 4). However, *p* values and confidence intervals were jointly reported together in relatively fewer abstracts, results-texts and discussions sections than any other combination of reporting elements that included *p* values.

## Detailed review

### *P values*

On most occasions, the precise magnitudes of *p* values were reported (Fig. 5A). Nevertheless, in at least 10% of all sections, including 44% of abstracts and 40% of discussions, all *p* values were written as inequalities (Fig. 5A). The odds of *p* values being written as inequalities in abstracts was substantially elevated (largely compatible with OR > 2) compared to in results-texts (OR = 3.2; 95% CI [1.3–7.6]) and -tables (OR = 5.2; 95% CI [2.0–13.9]) but only slightly so compared to in discussions (OR = 1.4; 95% CI [0.44–4.3]). The confidence interval in this latter case suggests that the results are also compatible with lower odds in abstracts (Fig. 6A).

Most reported *p* values were less than 0.05. This was particularly so in abstracts (88%) but also true in results-texts (65%) and -tables (60%) (Fig. 5B). In sections where *p* values
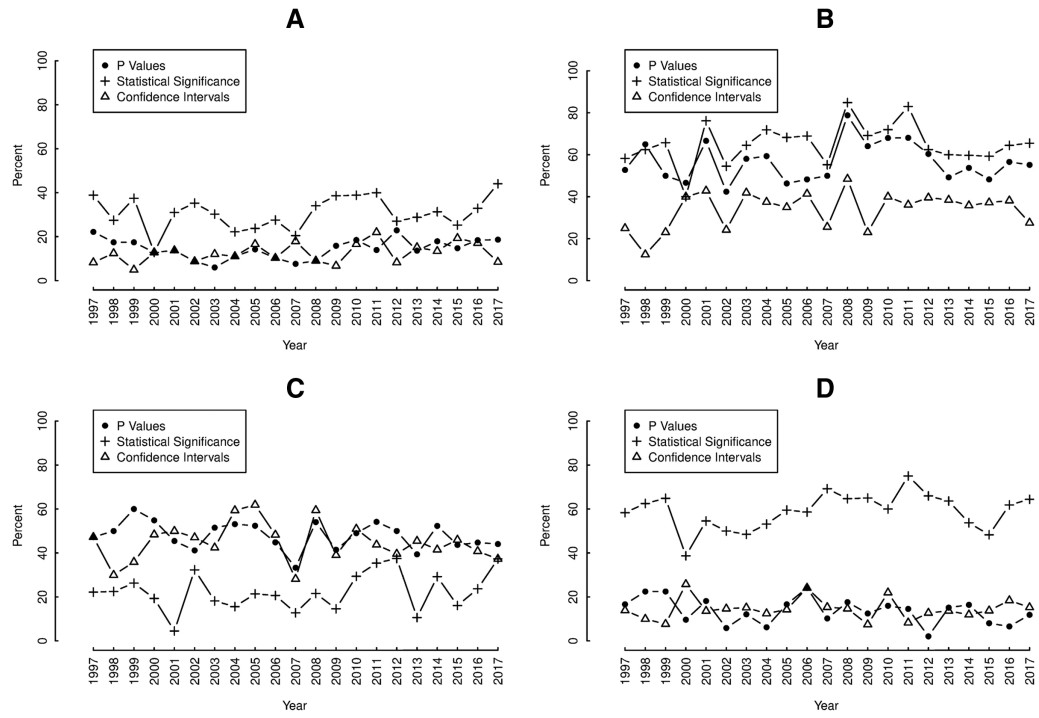

**Figure 3 Annual proportions of (A) abstracts, (B) results-texts, (C) results-tables and (D) discussion sections reporting $p$ values (•), confidence intervals (Δ) and statistical significance (+).** Plots of the yearly proportions (%) of (A) abstracts, (B) results-texts, (C) results-tables and (D) discussion sections reporting $p$ values (•), confidence intervals (Δ) and statistical significance (+) in original research articles published in Preventive Veterinary Medicine (1997–2017). In each abstract, result-text, results-table and discussion "$p$", "CI C.I. CL C.L. confidence" and "significance significant significantly" indicated the presence of $p$ values, confidence intervals and statistical significance, respectively.

were reported, the odds of a $p$ value less than 0.05 being reported in an abstract was substantially greater (consistent with OR > 3) than in results-texts (OR = 4.5; 95% CI [2.3–9.0]) and -tables (OR = 7.5; 95% CI [3.8–14.8]) while comparing results-texts to -tables, the results were mostly compatible with just slight (1 < OR < 2) increases (OR = 1.7; 95% CI [1.4–2.0]) (Fig. 6B). Overall, the proportion of $p$ values less than 0.05 showed no net tendency towards increase or decrease over the study period (Fig. 5C).

## Confidence intervals

In sections where confidence intervals were reported, most were not interpreted (Fig. 7A). In a higher proportion of discussions (61%) compared to abstracts (38%) and results-texts (29%), some or all of the reported confidence intervals were interpreted in a manner synonymous with significance tests (Fig. 7A). In discussions, the odds of at least one of the reported confidence intervals being interpreted like a significance test was substantially higher compared to in abstracts (OR = 3.2; 95% CI [0.8–13.5]) and even more so when compared to results-texts (OR = 4.9; 95% CI [1.3–18.2]). However, both estimates are imprecise and the confidence intervals indicate that these results are also compatible with slight associations, though less so than with strong ones (Fig. 6C). Most abstracts (93%;

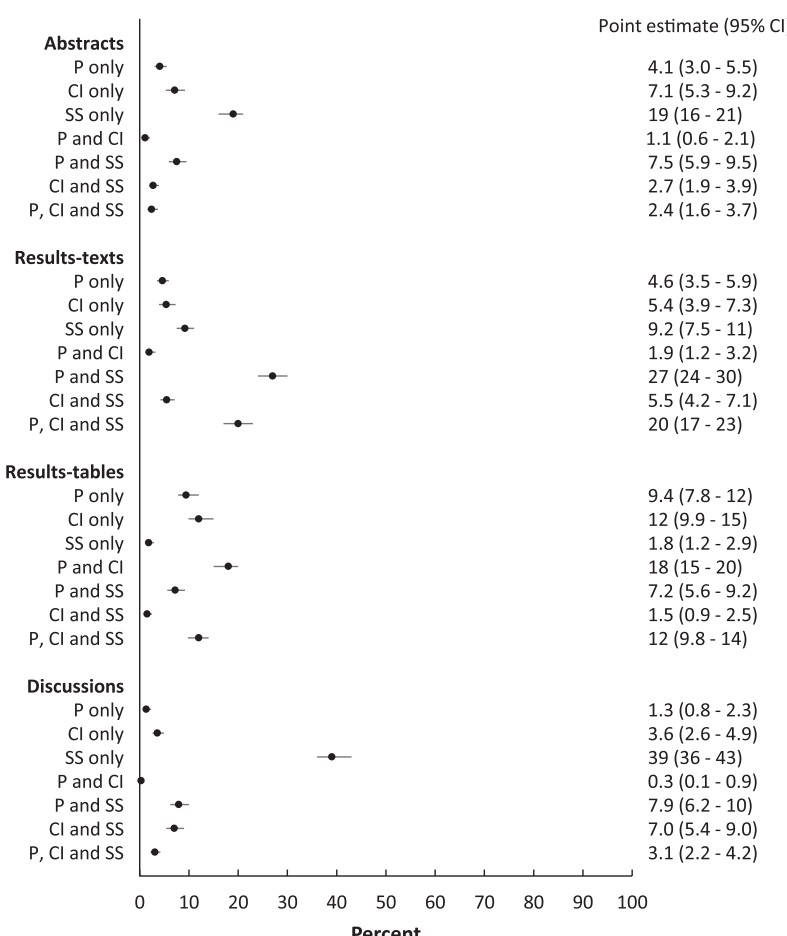

**Figure 4 Proportions of abstracts, results-texts, results-tables and discussion sections reporting combinations of *p* values, confidence intervals and statistical significance.** Point estimates and 95% confidence intervals for the proportion (%) of abstracts, results-texts, results-tables and discussion sections reporting various combinations of *p* values (P), confidence intervals (CI) and statistical significance (SS) in original research articles published in Preventive Veterinary Medicine (1997–2017).

95% CI [77–98%]) and results-texts (61%; 95% CI [48–72%]) reporting confidence intervals, reported them for only commonly used epidemiologic measures (Fig. 7B) not for other measures.

## Statistical significance

In sections where "significant…" referred to study results, most times this pertained to only statistically significant (63%; 95% CI [56–69%]), as opposed to only statistically non-significant (14%; 95% CI [9–19%]) results. This was true among both abstracts (81%; 95% CI [70–88%] *vs*. 7%; 95% CI [3–16%]) and discussions (52%; 95% CI [43–61%] *vs*. 34%; 95% CI [25–43%]) (Fig. 7C). The odds of "significant…" being used to refer to only statistically significant results was substantially higher (OR > 2) in abstracts than in discussions (OR = 4.1; 95% CI [1.9–8.6]) (Fig. 6D).

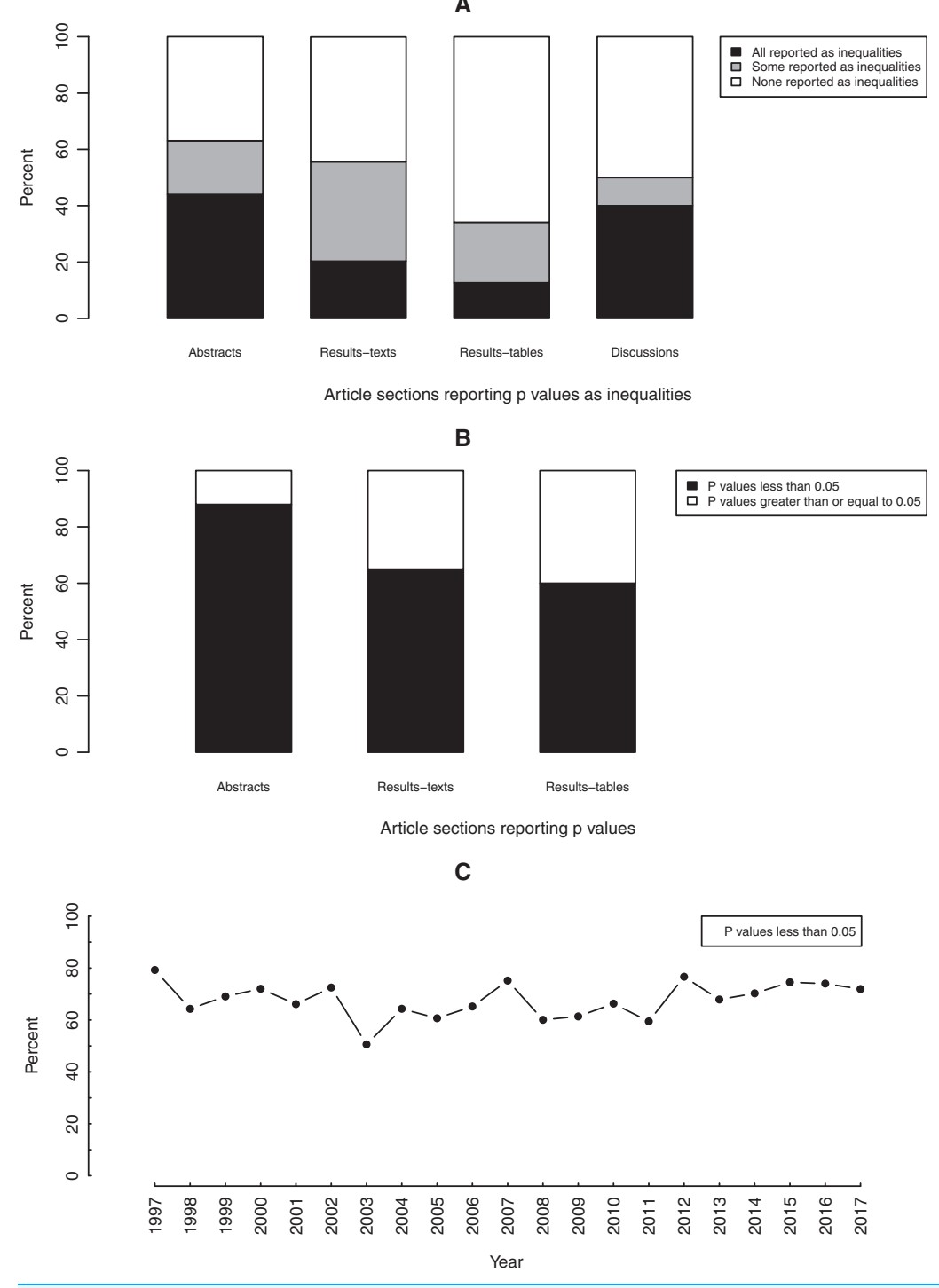

**Figure 5 Graphs depicting the reporting of *p* values.** Bar graphs showing (A) the proportions of article sections reporting *p* values as inequalities and (B) the proportions of *p* values less than 0.05 in each article section. Line graph (C) showing the total yearly proportion of *p* values less than 0.05 reported over the study period, in original research articles published in Preventive Veterinary Medicine (1997–2017). All proportions expressed as percentages.

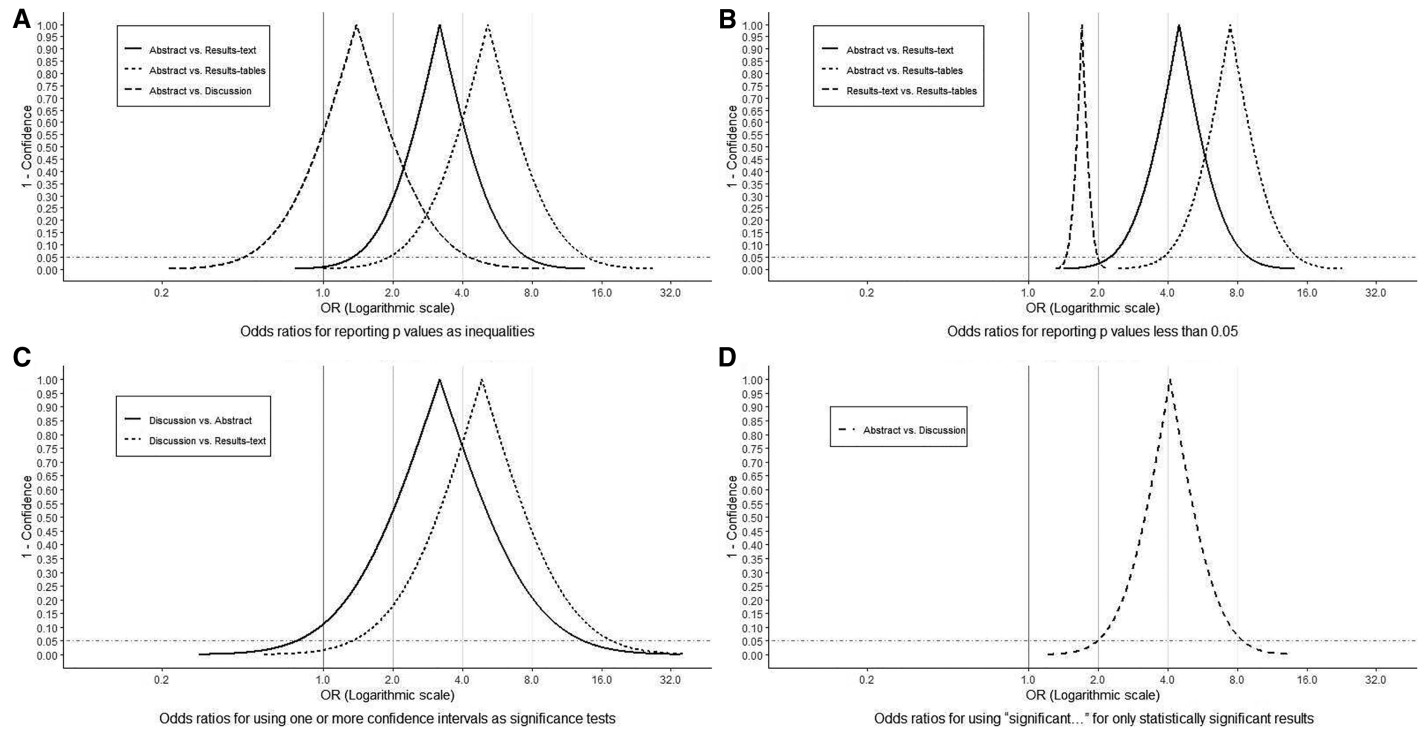

**Figure 6 Odds Ratio confidence interval functions.** Odds Ratio (OR) confidence interval functions for (A) reporting $p$ values as inequalities, (B) reporting $p$ values less than 0.05, (C) using one or more confidence intervals as significant tests and (D) using "significant…" for only statistically significant results. The peak of the confidence interval function represents the point estimate, while its first and second intercepts with the horizontal dotted line, represent lower and upper 95% confidence limits, respectively. All OR values on the horizontal axis corresponding to points on the curve are compatible within the data. The proportion of the area under the curve to the right of the vertical lines illustrates the level of compatibility of the observed associations with parameter values of at least those magnitudes (OR = 1, 2, 4 and 8).

# DISCUSSION

It has been suggested that the scientific community overvalues the importance of NHST to the interpretation of data (*Rothman, 2014*). One consequence of this misunderstanding is the frequent use of statistical significance, in the quantitative sciences, as the yardstick for judging a manuscript's worth. This in turn has led to concerns about the reproducibility and credibility of findings drawn from scientific research (*Wasserstein & Lazar, 2016*; *Lash, 2017*). Nevertheless, it is well recognized, among statisticians, that a result's $p$ value (statistical significance or lack thereof) is not synonymous with its scientific, clinical, practical or other subject matter-related significance (*Goodman, 2008*; *Wasserstein & Lazar, 2016*).

Our analysis of original research articles in Preventive Veterinary Medicine over a 20-year period, documents that statistical significance is reported in greater frequency than $p$ values and confidence intervals in all article sections except results-tables. Additionally, $p$ values are reported in similar frequency to confidence intervals in all article sections except results-texts, in which they (*i.e.*, $p$ values) are reported more frequently. These findings add further justification to the ASA's concern for the conduct and interpretation of data analysis in the quantitative sciences, as each article section reported overt indicators of NHST (*i.e.*, statistical significance and/or $p$ values) in higher frequencies than

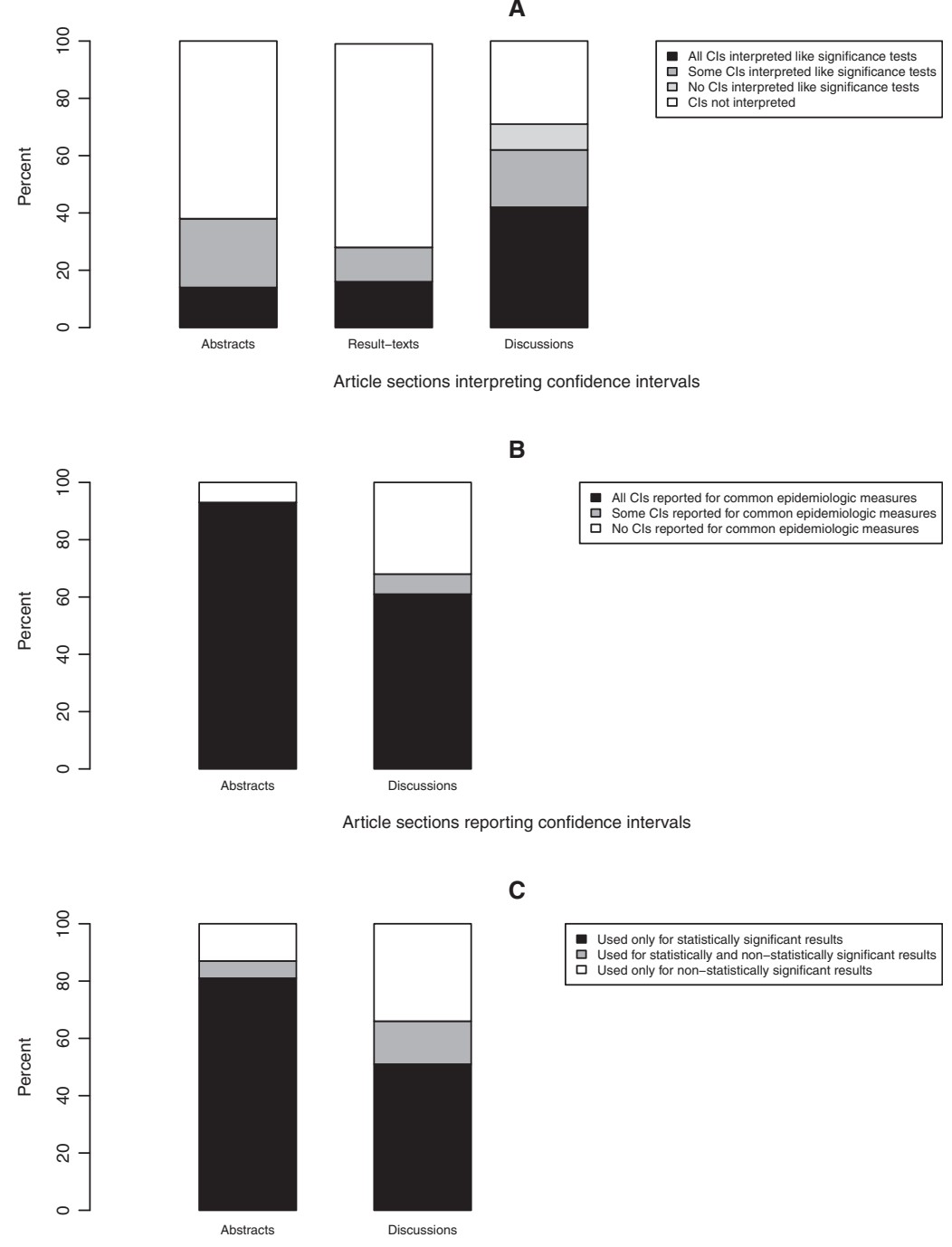

**Figure 7 Bar graphs depicting the reporting of confidence intervals (A and B) and statistical significance (C).** Bar graphs showing (A) the proportion of article sections in which confidence intervals (CIs) were interpreted, (B) the proportion of article sections reporting confidence intervals (CIs) and (C) the proportion of article sections using the term "significance", "significant" or "significantly" to report statistical significance in original research articles published in Preventive Veterinary Medicine (1997–2017). All proportions expressed as percentages.

confidence intervals. The lower observed frequency of reporting statistical significance (compared to $p$ values and confidence intervals) in results-tables appears natural, viewed against the backdrop that tables are dedicated to the presentation of numbers. However, the finding that approximately one in five tables (Fig. 1) report statistical significance (a qualitative categorization) and that the majority of these instances pertain to only statistically significant results, should also be seen as evidence of an overemphasis on NHST. This practice of reducing quantitative results into significant, non-significant dichotomies by authors has long been considered misleading by epidemiologists (*Rothman, 1978*). It not only diverts attention away from the magnitude of the parameter estimates in question but, particularly in tables, discourages critical examination of all the results and defeats the purpose of tabular presentation itself. More importantly, because it renders confidence intervals (and hence any consideration of the precision of estimates) "redundant", this habit can lead to erroneous conclusions of "no effect" and "conflicting results" in studies, in which the opposite is true (*Altman, 2000*; *Frieman et al., 1978*; *Powell-Tuck et al., 1986*; *Schmidt & Rothman, 2014*). One potential solution to this is for authors to develop an appreciation for both the location and precision of estimates, by learning to visualise confidence interval functions (*Lang, Rothman & Cann, 1998*; *Poole, 1987*; *Schmidt & Rothman, 2014*; *Sullivan & Foster, 1990*). Confidence interval functions (also called $p$ value functions) depict the entire range of parameter values and their degree of compatibility with the observed data at each level of confidence. This obviates the need to focus on a single arbitrary confidence level for inference. Any single point estimate and confidence interval is sufficient to generate the respective confidence interval function which can then be used for making inferences based on comparisons between different estimates and between estimates and effect sizes of practical importance. For instance, in Fig. 6C considering the spectrum of parameter OR values possible, it is clear that at all levels of confidence, both relationships depicted (discussions *vs.* abstracts and discussions *vs.* results-texts) are substantially more compatible with ORs > 2 than ORs < 2 and that both point estimates are similarly imprecise. We can verify this latter by calculating the ratio of upper to lower confidence limits at any confidence level (*Poole, 2001*) and comparing them (using the 95% confidence level, we get 17 and 14 respectively). These (most) important findings would not be noticed were just $p > 0.05$ (discussions *vs* abstracts) and $p < 0.05$ (discussions *vs* results-texts) reported. We refer the reader to a recently published tutorial on confidence interval function construction and use (*Infanger & Schmidt-Trucksäss, 2019*).

Our finding that no section reported confidence intervals with greater frequency than $p$ values, is contrary to a recent report of several medical and epidemiology journals which found that confidence intervals are now reported more frequently than all forms of NHST in the abstracts of all but three of them (*Stang et al., 2017*). It is also contrary to earlier findings from the American Journal of Epidemiology which report that by 1990, the proportion of results-tables reporting confidence intervals exceeded the proportion reporting the results of statistical tests (*Savitz, Tolo & Poole, 1994*). However, our finding that there is a tendency to interpret confidence intervals as if they were significance tests is in agreement with the findings of *Savitz, Tolo & Poole (1994)*. This practice, a covert
form of NHST, again reduces results to significant, non-significant dichotomies and detracts from gauging the precision of estimates for which the confidence interval is useful. Some credence for this view is provided by the finding that if a confidence interval was not interpreted like a significance test, most times it was not interpreted at all (Fig. 7A). The greater tendency for confidence intervals to be used like significance tests in discussions than in abstracts might reflect the fact that statistical significance can be communicated more efficiently *via p* values or "significant…" in abstracts, where there are word count restrictions.

The lack of secular trends in the proportions of sections containing the reporting elements over the study period, suggests that reporting practices in Preventive Veterinary Medicine have not improved over time. This is unexpected, given the sustained criticism of NHST by epidemiologists during the study period (*Greenland et al., 2016*; *Lang, Rothman & Cann, 1998*; *Poole, 2001*; *Rothman, Greenland & Lash, 2008*; *Stang, Poole & Kuss, 2010*) and also contrasts with results from the aforementioned studies of reporting in epidemiology journals where improvements were observed (*Savitz, Tolo & Poole, 1994*; *Stang et al., 2017*). In particular, it was found that in abstracts, the prevalence of confidence intervals increased while NHST decreased in all but one journal studied (1975–2014) (*Stang et al., 2017*) and the use of confidence intervals in both results-texts and -tables increased from 1970 to 1990 (*Savitz, Tolo & Poole, 1994*). Reports on other journals publishing biomedical literature, have found increases over time in the presentation of *p* values in the abstracts (*Chavalarias et al., 2016*), tables and figures (*Cristea & Ioannidis, 2018*).

Our results also show that the presentation of *p* values as inequalities (usually "$p < 0.05$") is frequent in abstracts and discussions (Fig. 5A). In abstracts, this might be in order to highlight findings considered most important to journal editors, reviewers and readers, as it is often the only article section freely accessible to everyone online (*Cals & Kotz, 2013*), the one read first and sometimes the only one read. Adding support to this view are the findings that a higher proportion of *p* values less than 0.05 were reported in abstracts than in results-texts and -tables and that the corresponding associations are both substantially compatible with OR > 2 and OR > 4, respectively (Fig. 6B). These findings suggest a tendency towards selective reporting of *p* values that don't exceed an arbitrary threshold and is further evidence of the use of statistical significance as a determinant of the importance of results in a manuscript. Reporting *p* values as inequalities is incorrect practice, as it says only that the corresponding observed difference or effect falls within a particular range (usually $p \geq 0.05$ or $p < 0.05$). This renders it impossible for the reader to determine the level of compatibility of the results with a given hypothesis and erroneously conveys the impression that all *p* values below any given threshold (e.g., 0.05) correspond to equal consistency with the null hypothesis (*Goodman, 2008*; *Greenland et al., 2016*).

This study has a number of limitations. We examined reporting practices for only one journal, which may prevent us from extrapolating our findings to other journals publishing veterinary research. However, given the widespread consensus on current misuse of NHST in the biomedical sciences, we thought it might be more beneficial to provide

evidence of this in a major veterinary epidemiologic journal. We did not document the use of Bayesian, likelihood or other legitimate methods of statistical inference. While this would have been informative, compared to NHST, these methods are less frequently used in veterinary epidemiology and their widespread misuse is not currently a concern in the biomedical literature. Finally, we did not perform animal species–specific analyses and so cannot comment on any differences between reporting practices of researchers focussing on production, as opposed to companion animals. This might be relevant, as anecdotally, most veterinarians with formal training in epidemiology, conduct research on production animals. Notwithstanding these limitations, we believe this work contributes uniquely to the veterinary medical research literature by describing, for the first time, reporting practices in a leading veterinary epidemiological journal and highlighting areas in which reporting practice lags behind those in other epidemiology sub-specialities.

## CONCLUSIONS

We summarise our findings with the following points regarding published articles in Preventive Veterinary Medicine: (1) Despite variation, the frequency of reporting of $p$ values, confidence intervals and statistical significance in articles has remained relatively stable over the last two decades. (2) Over this period, confidence intervals have been reported less frequently than either $p$ values or statistical significance. (3) When interpreting confidence intervals, on most occasions, authors use them as surrogates for hypothesis tests. (4) Statistical significance is reported by authors in greater frequency than $p$ values or confidence intervals in all sections except results-tables and observed patterns of NHST reporting suggest that authors preferentially use the terms "significant", "significance" or "significantly" to report statistically significant as opposed to statistically non-significant results. (5) Authors often report $p$ values using inequalities, preferentially report those with magnitudes less than 0.05 and primarily do so in the abstracts of manuscripts.

It is hoped that this study will be useful to the field of veterinary epidemiology in providing an indication of both recent historical, and current reporting practice with regard to these elements and in identifying where it stands with regard to null hypothesis significance testing within the wider discipline of epidemiology. We also hope it provides insight to editors, reviewers and authors in this field, about areas in which reporting might improve.

## ACKNOWLEDGEMENTS

The authors thank Farla Kaufmann, Mary Codd and Brant Schumaker for helpful comments on previous drafts.

### Funding

The authors received no funding for this work.

## Competing Interests

The authors declare that they have no competing interests.

## Author Contributions

- Locksley L. McV. Messam conceived and designed the study, conducted the study, analyzed the data, prepared figures and/or tables, authored or reviewed drafts of the paper, and approved the final draft.
- Hsin-Yi Weng conceived and designed the study, conducted the study, analyzed the data, prepared figures and/or tables, authored or reviewed drafts of the paper, and approved the final draft.
- Nicole W. Y. Rosenberger conducted the study, analyzed the data, authored or reviewed drafts of the paper, and approved the final draft.
- Zhi Hao Tan conducted the study, analyzed the data, authored or reviewed drafts of the paper, and approved the final draft.
- Stephanie D. M. Payet conducted the study, analyzed the data, authored or reviewed drafts of the paper, and approved the final draft.
- Mahishi Santbakshsing conducted the study, analyzed the data, authored or reviewed drafts of the paper, and approved the final draft.

## Data Availability

The raw data is available in the Supplemental File.

## Supplemental Information

Supplemental information for this article can be found online at http://dx.doi.org/10.7717/peerj.12453#supplemental-information.

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
