# Peer review of "The reporting of p values, confidence intervals and statistical significance in Preventive Veterinary Medicine (1997–2017)"

_PeerJ, doi:10.7717/peerj.12453_

## Round 0.1 · original submission · Major Revisions

Experts have reviewed your study and found merit in its aims and methods and identified instrumental feedback and recommendations. Most important include the justification for this study and for the use of CIs as an alternative; that despite its validity, the manuscript should provide more explanations both for reasoning and demonstration for the sake of the reader.

·

Basic reporting

I thought the basic reporting was fine, though the review of previous work could have been more extensive. For example, the author said nothing about the banning of p-values in Basic and Applied Social Psychology in 2015.

Experimental design

I thought the sampling was fine, though limited. It isn’t clear to me why more journals were not sampled, which would have increased the generalizability of the findings. It also would have been nice to have had more explanation for the reasons that particular choices were made.

Validity of the findings

I thought there was a strangeness in the reporting of the findings. It is obvious that the authors do not favor p-values, and they are right in this. And I understand why the authors have some of their focus on using confidence intervals (CIs) as significance tests; they are right to be critical of this, and it matters, as I will explain in the general comments. But the authors fail to explain why their use of CIs is valid! I see this as a major failing, as I question that. What EXACTLY is the reader to conclude from a CI? Should the reader conclude that the population parameter of interest has a 95% chance of being within the CI? Of course not, that would be blatantly invalid. But if the reader is not to draw that invalid conclusion, then what, precisely, should the reader conclude? The point I am making is that the author’s use of CIs also is invalid. The authors should either (a) explain exactly how they are using CIs validly (which I don’t think they can do without watering down the gain to the point of triviality) or (b) dump the CIs. They could, for example, use the a priori procedure (by yours truly) to say how precise their data are.

·

Basic reporting

The paper is well-written, and relatively easy to read. The section with the results is a bit dense (quite a lot of numbers to process here), but I think the outcomes that are mentioned are relevant, so there's little that can be done. As is often the case, I found the figures easier to process than the written outcomes.

Although I'm generally quite positive about the paper, one of my main concerns deals with the amount of background information (see also general comments). Although I don't have to be convinced of the relevance of the research question, I think it might be good to add a bit more context to the reader who might be less familiar with issues surrounding inference. The issues are briefly mentioned, but personally I'd prefer a bit more context here: What is the problem with focussing on a binary outcome? For what problem would CIs be a solution? And, to be fair and balanced, are CIs without problems? Is there something wrong with p-values presented as inequalities? Long story short, I think it's important to link the research questions to background information. Thus, the paper would not only present some outcomes about the (lack of) change in the reporting of inferential results, but at the same time make the reader familiar with some often debated issues in statistical literature.

Experimental design

I think the study is well-designed, so I have few comments about that. I was expecting an extensive analysis for the change of reporting practices over time, until I saw the Figures with the results, which immediately made clear that such an extensive analysis would be superfluous. Moreover, I appreciate the amount of detail of the Method, and the supplementary materials.

Validity of the findings

The results are interpreted quite carefully, and I've seen no clear cases of exaggeration or overgeneralization.

Additional comments

Review
First of all, I appreciate to have been given the opportunity to review this paper. In 2006, we’ve published a similar study on the use of p-values and CIs the psychology literature, and it is nice to see that such studies are also carried out in different fields. I also want to make clear that my expertise is in the use of statistical methods, but not in epidemiology or veterinary medicine, but for this paper that doesn’t matter too much, apart from some the fact that if I’d written this paper, I’ve leaned more on literature based on the reporting of inference in the social sciences.
I think this is quite a good paper, and I found it interesting to read. Actually, I have only one larger concern, and a few (very) minor issues. My main concern is that I think that more background information for this study would be helpful to understand why the study was conducted, and could be educational for the reader as well. To be clear, *I* don’t have to be convinced of the importance of such a study, but I’m not sure whether a more substantively oriented person in this particular field is aware of all the discussions about how to report inferential results. Strengthening of the introduction could alleviate that issue. Of course, readers could read relevant and mentioned literature like Wasserstein and Lazar (2016) themselves, but I’d be in favour of giving a bit more context to the reader. Questions I’d like to have seen addressed are: What exactly is criticized about the reporting of inferential results? What’s wrong with p-values, or with a binary interpretation of them? Why are CIs seen as a useful alternative, and are they without criticism themselves? [to be frank, I’ve been involved in some papers on the problems with the interpretation of CIs, so this remark may not be completely neutral. I do think though that we have to be careful in saying that the interpretation of p-values is often problematic (and it is!), and that therefore moving to CIs solves all problems] What’s wrong with interpretating a CI as if it were a significance test, given their technical relationship? Why would one not report a p-value as an inequality? What kind of a change over time could have been expected? Would one expect CIs for both common and not so common methods (and do they even exist for the latter)? All these issues come back to some extent in the results, but are not thoroughly discussed in the introduction. I think these could be better matched, with, as mentioned before, the additional benefit of making the paper more informative/educational. In case it wasn’t clear yet: By no means am I suggestion that the authors wouldn’t have good answers to these questions: I’m sure they have. I’d just like them to be more explicit about them. A related, but arguably less urgent point deals with the difference between NHST and significance testing (e.g., line 401). I’m pretty sure this will go over the heads of many readers, so I’d either leave this out, or explain the distinction a bit more.
Another, somewhat smaller issue, is that I think a rationale for focussing on p-values and CIs only is lacking. Although I can easily imagine a good justification (their frequency of use, or how often they are discussed in literature, for example), the paper just states that the focus will be on them. Why not include Bayesian statistics, for example? I’m not saying the authors should have done that, but I’d like to see a justification for their selection added,
I also have a few (very) minor issues:
94: While it may be postulated,  I think the comma is redundant, but then again, I’m no native speaker
394: p ≥.05 or p < 0.05. It’s not important at all, but I’d suggest to be consisted in whether or not a “0” proceeds a decimal number
Figure 1: I have no strong opinion about this, but I’m not sure whether this figure contains a lot of information. To me, it wasn’t hard to imagine how such a search process would look like, but maybe I’m biased because of my familiarity with such studies.
Figure 4: I think it would be helpful if a legend would be added to the figure. Of course, there were only three elements to remember, but generally I think it is easier to read a figure, rather than having to go to the text instead. Of course, this is also due to the incomprehensible habit of some publishers to present the text and the figures on separate pages: I would be so much easier if they would be presented in the same manner as they would in the eventual published paper.

To summarize, I think this is a decent paper, with the potential to be a good paper if the outcomes presented in the results section would have had a better link to issues discussed in the introduction.

I always sign my reviews.
Rink Hoekstra

---

## Round 0.2 · accepted · Accept

Thank you for responding to the reviewers' comments, they received them positively suggesting only minor notes on furthering the explanation for confidence intervals versus p values, however, after studying the revised manuscript I believe you have addressed their most recent set of comments and that the manuscript is ready for publication.

Congratulations and best wishes in your future research!

·

Basic reporting

Fine

Experimental design

Fine

Validity of the findings

Fine

Additional comments

Well both myself and Rink Hoekstra commented on the confidence interval issue in the first round of reviews. I’m not really satisfied with the author’s response to our comments on that. However, I’d hate to see that disagreement scuttle the paper. Therefore, despite my lack of satisfaction on that issue, I’ll recommend publication. I hope that it influences the field away from p values.

·

Basic reporting

I have nothing to add to my previous remarks: this is all fine

Experimental design

Same here

Validity of the findings

Same here.

Additional comments

I appreciate the opportunity given to review this paper a second time. I think the authors did a good job in processing the comments by both myself and David Trafimow, and I think most issues have been dealt with properly.
I do want to note that the authors are clearly more optimistic about the usefulness of CIs than I am, which may come as now surprise given our papers on this issue. As Trafimow also pointed out, too often CIs are promoted as a valuable alternative for NHST, without explaining what one can validly conclude from a given CI. This is actually quite tricky, and makes, at least in my opinion, the use of CIs less attractive than many would think (and than I used to believe for quite a while as well). The answer given to Trafimow’s comment in the reply (point 3) is not completely convincing to me: “reasonably compatible” is not formally defined as far as I know, and I wouldn’t know how to use that, unless as an inverted test. Moreover, it suggest that values outside the interval are reasonably incompatible, which I’m not sure I agree with either, and it notably creates a dichotomy that CIs are often propagated to avoid.
But then again, I do agree that a relatively careful use of CIs is probably preferable over what we often see in substantive studies (as this study shows as well), so overall I think the added value of publishing this paper outweighs the fact that I do not fully agree with the authors’ assessment of CIs.
Best regards,
Rink Hoekstra